# TTT (Tel2-Tti1-Tti2) Complex, the Co-Chaperone of PIKKs and a Potential Target for Cancer Chemotherapy

**DOI:** 10.3390/ijms24098268

**Published:** 2023-05-05

**Authors:** Sankhadip Bhadra, Yong-jie Xu

**Affiliations:** Department of Pharmacology and Toxicology, Boonshoft School of Medicine, Wright State University, Dayton, OH 45435, USA

**Keywords:** TTT, Tel2, Tel2-Tti1-Tti2, PIKKs, ATR, ATM, DNA-PKcs, mTOR, SMG1, TRRAP, Hsp90, R2TP, Asa1, Cdc37, protein kinase, co-chaperone, ivermectin

## Abstract

The heterotrimeric Tel2-Tti1-Tti2 or TTT complex is essential for cell viability and highly observed in eukaryotes. As the co-chaperone of ATR, ATM, DNA-PKcs, mTOR, SMG1, and TRRAP, the phosphatidylinositol 3-kinase-related kinases (PIKKs) and a group of large proteins of 300–500 kDa, the TTT plays crucial roles in genome stability, cell proliferation, telomere maintenance, and aging. Most of the protein kinases in the kinome are targeted by co-chaperone Cdc37 for proper folding and stability. Like Cdc37, accumulating evidence has established the mechanism by which the TTT interacts with chaperone Hsp90 via R2TP (Rvb1-Rvb2-Tah1-Pih1) complex or other proteins for co-translational maturation of the PIKKs. Recent structural studies have revealed the α-solenoid structure of the TTT and its interactions with the R2TP complex, which shed new light on the co-chaperone mechanism and provide new research opportunities. A series of mutations of the TTT have been identified that cause disease syndrome with neurodevelopmental defects, and misregulation of the TTT has been shown to contribute to myeloma, colorectal, and non-small-cell lung cancers. Surprisingly, Tel2 in the TTT complex has recently been found to be a target of ivermectin, an antiparasitic drug that has been used by millions of patients. This discovery provides mechanistic insight into the anti-cancer effect of ivermectin and thus promotes the repurposing of this Nobel-prize-winning medicine for cancer chemotherapy. Here, we briefly review the discovery of the TTT complex, discuss the recent studies, and describe the perspectives for future investigation.

## 1. Introduction

The human genome encodes 518, the fission yeast *S. pombe* expresses 106, and the budding yeast *S. cerevisiae* possesses 120 protein kinases [1,2,3] that regulate cellular responses to growth factors, environmental signals, and internal processes by the regulation of protein interactions, enzyme activity, or protein localization. Among the protein kinases, PIKKs (phosphatidylinositol 3-kinase-related kinases) are a group of proteins that are structurally related to lipid kinase but possess protein kinase activities [4,5]. They phosphorylate serine or threonine residues on their target proteins in their phospho-signaling pathways. Although TRRAP (transformation/transcription domain associated protein), a protein of multiple chromatin complexes for transcription activation, does not have the kinase activity of other PIKKs, the overall structural domains of all PIKKs are quite similar. Some PIKKs are conserved from yeasts to humans that play important roles in genome maintenance and cell proliferation. The majority of the protein kinases in the kinome require the co-chaperone Cdc37 that brings the client kinases to the heat shock protein Hsp90 for proper folding, stability, and biogenesis [6,7,8]. Similar to Cdc37, the TTT complex has been an established co-chaperone that works with Hsp90, specifically for the maturation and stability of PIKKs. This function of the TTT and the potential mechanistic pathways involved have been extensively reviewed by Sugimoto 5 years ago [9]. Recently, the TTT complex has been shown to work co-translationally on the newly synthesized peptides of PIKKs to promote their maturation and biogenesis [10]. Structural studies have revealed the TTT structure as a single entity or in complex with the R2TP (Rvb1-Rvb2-Tah1-Pih1) [11,12,13], the complex that interacts with Hsp90. A series of mutations have been identified in the TTT that cause the neurodevelopmental syndrome characterized by intellectual disability, global developmental delay, microcephaly, abnormal movements, and various other pathogenic phenotypes [14,15,16,17]. Very recently, Tel2 in the TTT complex has been found to be a target of ivermectin (IVM) [18], an antiparasitic drug that has been successfully used in clinics during the past decades. This discovery explains the antitumor effect of IVM and thus provides the mechanistic foundation for repurposing this clinically established drug for cancer chemotherapy. Altogether, these advances will likely drive a new wave of research interest in the TTT complex. In this review, we briefly describe the discovery of the TTT, review the recent studies, and discuss a few questions that can be addressed in the future. 

## 2. Discovery of the TTT Complex and Its Homologs

Tel2 is the first subunit discovered in the TTT complex. It was first reported more than 40 years ago in the nematode *C. elegans* as a *rad-5* mutant with hypersensitivity to ionizing radiation [19] and in the budding yeast *S. cerevisiae* as a *tel2-1* mutant with a short telomere phenotype [20]. The *tel2-1*/*rad-5* was rediscovered as *clk-2* in *C. elegans* by screening for genes involved in biological rhythms and lifespan [21,22]. Later, it was found that *rad-5* is allelic with *clk-2* [23]. While *rad-5* carries a missense mutation resulting in G135C amino acid change, the mutation in *clk-2* causes C772Y substitution. Both mutants show temperature sensitivity (*ts*), embryonic lethality, and defects in the DNA damage and the replication checkpoint pathways [23,24,25]. An early study showed that although the RAD-5/CLK-2 protein is homologous to the budding yeast Tel2 and conserved in eukaryotes [23], the two missense mutations identified have little effect on telomere length. However, later studies have shown that CLK-2 has a function in telomere length regulation [26,27]. The budding yeast *tel2-1* is a *ts* mutant, consistent with its essential function for cell survival [28]. It shows the short telomere phenotype only after ~150 generations [20]. The *tel2-1* mutant carries a missense mutation that causes an S129N substitution in Tel2 [28]. An *in vitro* study has shown that the Tel2 protein binds telomeric double-stranded DNA [29]. However, unlike the *rad-5/clk-2* mutants that are sensitive to DNA damage and the replication stress induced by hydroxyurea (HU), the *tel2-1* mutant is insensitive to HU, although the mutation interrupts its interactions with Tel1 and Mec1, the ATM, and ATR homologs, respectively, in budding yeast [30,31]. Studies in mammalian cells have shown that CLK-2(Tel2) plays an important role in the ATR checkpoint signaling pathway [27,32,33,34]. In the fission yeast *S. pombe*, the Rad3(ATR) signaling in the replication checkpoint is eliminated in the *tel2-C307Y* mutant [35] or becomes defective in cells with acute Tel2-depletion [36]. Co-immunoprecipitation of Tel2 followed by LC-MS/MS analysis in *S. pombe* uncovered two Tel2 interacting proteins Tti1 and Tti2, which are all conserved in eukaryotes [37,38,39]. These studies also discovered that Tel2 interacts with all PIKKs in fission yeast [37,38]. The TTT complex was subsequently confirmed by the proteomic studies in budding yeast and fission yeast [39] and the genome-wide RNAi screen in mammalian cells [40] (see Table 1).

## 3. Tel2 Deletion Reduces the Protein Levels of All PIKKs

The pleiotropic phenotypes of the non-lethal mutants in yeasts and *C. elegans* suggest that Tel2 works in distinctive cellular processes. Sequence analysis of Tel2, however, did not identify any defined domain in the protein except helical repeats [30], which raises a question of why its mutations have such a broad and profound impact in biology. This long-standing question was finally solved by Takai et al. in 2007 [41]. They found that while deletion of the *tel2* gene is embryonically lethal in mice, its conditional deletion in embryonic fibroblasts compromises checkpoint response to ionizing radiation. More importantly, these defects of *tel2* deletion corroborated with the reduced levels of proteins, not mRNAs, of all six mammalian PIKKs (Table 1). Since PIKKs are signal transducers that dictate multiple signaling pathways, this result shows that the multiple biological functions of Tel2 are realized by promoting the stability of the PIKKs. Subsequent studies in yeasts [30,31,35,42,43,44,45,46] and mammalian cells [13,40,47,48,49] showed that point mutations of Tel2 or the depletion of any subunit of the TTT reduced the levels of PIKKs, which confirm the result.

As shown in Figure 1, PIKKs are a group of large proteins of 270–450 kDa and signal transducers (see reviews [5,50,51,52,53,54]). PIKKs phosphorylate serine or threonine residues at the target proteins using the C-terminal protein kinase domain related to the PI3 kinase. The kinase domain, accounting for only 5–10% of the total sequence, is flanked by the FAT (FRAP-ATM-TRRAP) and FATC (FAT C-terminus) domains, and most of their N-termini are composed of long arrays of HEAT (huntingtin, elongation factor 3, protein phosphatase 2A and TOR1) repeats [55]. Among the six PIKKs, ATM (ataxia-telangiectasia mutated) and ATR (ATM and Rad3-related protein) are conserved in all eukaryotes that govern the response to DNA damage or replication stress by phosphorylating proteins involved in DNA repair and cell cycle regulation. Although partially redundant, ATM is activated mainly in response to DNA double-strand breaks, whereas ATR is activated by single-strand DNA generated at the DNA damage sites or stalled replication forks. In mammals, there are four additional PIKKs: DNA-PKcs, mTOR, SMG1, and the catalytically inactive TRRAP. DNA-PKcs is the catalytic subunit of DNA-dependent protein kinase. It is crucial for DNA repair of double-strand breaks by non-homologous end-joining. SMG1 (suppressor with morphological effect on genitalia 1) regulates nonsense-mediated decay of aberrant mRNA. The mammalian target of rapamycin (mTOR) is the catalytic subunit of TORC1 and TORC2 complexes. It controls cell growth in response to nutrient availability, mitogenic signals, and environmental cues. TRRAP regulates gene expression via its association with histone acetyltransferase complexes. Because the TTT is required for the stability of all PIKKs, it is essential for cell viability, and its mutations cause the pleiotropic phenotypes mentioned above.

## 4. TTT Functions as a Co-Chaperone for the Stabilization of PIKKs

Although the complex phenotypes of the TTT mutants have been reconciled by the ability to promote the stability of PIKKs, how the stability is achieved by the TTT remained unknown because all three subunits are predicted to contain largely the HEAT repeats [13,30,49]. Using peptide arrays as the substrates, Horejsi et al. discovered in 2010 that human Tel2 is phosphorylated by Casein Kinase 2 (CK2) at serine^487^ and serine^491^ in the middle of the protein [60]. Using phospho-peptides to pull down the Tel2 interacting proteins, they uncovered the R2TP complex and its associated prefoldin-like complex. Within the R2TP complex, Pih1 (protein interacting with Hsp90) directly binds to the phosphorylated Tel2 as well as the Tah1 (TPR-containing protein associated with Hsp90) subunit. The TPR (tetratricopeptide repeat) domain of Tah1 binds to the conserved MEEVD motif in the C-terminal tail of Hsp90 [61,62]. Therefore, R2TP interacts with the TTT and Hsp90 [48,63], a chaperone that is known to be regulated by a set of co-chaperones [64,65]. This indicates that TTT acts as a scaffold to coordinate the activities of R2TP and Hsp90 for the stability of PIKKs. Consistent with this mechanism, the CK2 phosphosite mutant of Tel2 retains the association with PIKKs but interrupts the interaction with R2TP, causing the instability of PIKKs, particularly mTOR and SMG1. 

Hsp90 is an ATP-dependent molecular chaperone that is required for cell viability and highly conserved from bacteria to mammals [66,67]. In mammals, Hsp90 is present in the cytoplasm (Hsp90α and Hsp90β), mitochondria (TRAP1), and endoplasmic reticulum (Grp94) [68,69,70]. Hsp90 collaborates with Hsp70 and a set of co-chaperones to stabilize and activate several hundred ‘client’ proteins and complexes. The co-chaperones recognize the clientele proteins and deliver them to the chaperone. The Hsp90 client proteins typically play essential roles in cell signaling and adaptive stress response. Hsp90 is a homodimer of dynamic conformations. It has a highly conserved N-terminal ATP-binding domain, a middle domain that binds client proteins and co-chaperones, and a C-terminal domain that dimerizes Hsp90. ATP binding and subsequent hydrolysis drive a cycle of conformational changes that is essential for the chaperone activity [71,72]. Two extreme conformational states, the tense and relaxed states, of the Hsp90 homodimer have been observed [72,73,74]. In the tense state, ATP binding associates the N-terminal domains, leading to the closed structure. In the relaxed state, ATP hydrolysis dissociates the N-terminal domains, leading to a V-shaped open state. Although Hsp90 is known to work with a set of co-chaperones, the best-understood and simplest co-chaperone is Cdc37 [75,76]. Cdc37 can also interrogate protein kinases by testing their ability to resist local unfolding [7,77], and acts as a gatekeeper to deliver the protein kinases to Hsp90 for proper folding, biogenesis, and activation. 

In addition to the PIKKs, the human genome encodes > 500 and the yeasts possess >100 protein kinases [1,2,3]. Most of the conventional kinases require Cdc37-Hsp90 for maturation and activation [6,65]. Although the TTT does not directly interact with Hsp90 like Cdc37 as the current model suggests, its function in promoting the stability of PIKKs is nonetheless reminiscent of that by Cdc37-Hsp90. CK2 is also a client protein kinase of Hsp90-Cdc37 [6,78,79,80]. It phosphorylates Cdc37 [80] as well as Tel2 and Tti1 in the TTT complex [47,60], and thus modulates the chaperone activity of Hsp90. CK2 is generally regarded as a constitutively active kinase. However, it can be regulated by various other protein kinases [81,82,83,84] and their interacting proteins or biomolecules [85,86,87,88,89,90,91]. Therefore, CK2 interconnects the TTT and Cdc37 and regulates their co-chaperone functions for the maturation and stability of most of the protein kinases in the kinome.

According to the current model, as a component of R2TP, Pih1 recruits Hsp90 to the TTT complex in a manner that depends on Tel2 phosphorylation by CK2. However, this mechanism is likely not highly conserved. First, the homologs of Tah1 and Pih1 remain to be identified in fission yeast and *C. elegans* (see Table 1). Second, although CK2 phosphorylation is observed at serine^490^ and serine^493^ in fission yeast Tel2 [92], the phosphosite mutation does not sensitize the cells to HU, DNA damaging agents, rapamycin, or other stresses, and nor does it affect Tel2 interactions with other proteins, including Hsp90. Finally, although the homologs of Tah1 and Pih1 exist in budding yeast, *pih1* depletion does not reduce the protein levels of Mec1(ATR) and Tel1(ATM) under physiological conditions [42]. Consistent with the *ts* phenotype of the *pih1∆* mutant, Mec1 and Tel1 become unstable when the temperature of the cell culture increases to 37 °C. Interestingly, Asa1, a highly conserved component of the chromatin remodeling ASTRA(ASsembly of Tel, RVB, and ATM-like kinase) complex [39], interacts with the TTT and stabilizes Mec1 and Tel1 under physiological conditions [42]. This shows that in budding yeast, the TTT can regulate the stability of PIKKs through at least two separate mechanisms: one through CK2 phosphorylation of Tel2 and the other via Asa1 [9]. 

## 5. TTT Structure and Its Interactions with Hsp90 and R2TP Complex

Purified human TTT is a stable complex of ~400 kDa and a heterotrimer with a 1:1:1 molar ratio [11,12]. TTT also forms a much larger and uncharacterized complex of ~2000 kDa in human cells [40]. Tel2 has a less conserved flexible linker of ~50 amino acids that connects its N-terminal domain (NTD) and C-terminal domain (CTD). The crystal structure of budding yeast Tel2 lacking the flexible linker shows that both the NTD and the CTD are composed of elongated helical HEAT repeats, which eventually fold into the α-solenoid structure in both the NTD and the CTD domains [13]. The two domains in Tel2 are organized in a nearly perpendicular orientation. Cryo-EM structures of the human TTT complex show that all three subunits are α-solenoid structures of the HEAT repeats [11,12]. The largest subunit Tti1 forms a curved structure that functions as the scaffold of the whole complex. Its central region interacts extensively with the NTD of Tel2 in an opposite direction, whereas its C-terminus binds to Tti2 [11,12]. Since the CTD of Tel2 does not bind to Tti1 [13], it is less visible [11] or even invisible [12] in the cryo-EM structures, although evidence suggests that it resides in a dynamic fashion near the interface between Tti1 and the NTD of Tel2 [11] (see Figure 2a). 

The mechanism described above suggests that CK2 phosphorylation of Tel2 in TTT recruits the TP (Tah1-Pih1) complex, which then recruits Hsp90 as the chaperone for PIKKs. Yeast Pih1 (344 amino acids) and Tah1 (111 amino acids) are two small proteins. Structural studies have shown that Pih1, also discovered as Nop17 [93], consists of an N-terminal PIH domain and a C-terminal CS domain that are interconnected by a flexible linker [94,95]. The N-terminal PIH domains bind proteins with the CK2 phosphorylation motifs such as Tel2. The C-terminal CS domain of Pih1 binds the extended C-terminal tail of Tah1. Tah1 contains an N-terminal TPR domain [61] that binds the conserved MEEVD C-terminal tail of Hsp90 [61,95]. Therefore, Hsp90 and the phosphorylated TTT are brought together by the TPR domain and the PIH domain of the TP complex formed by the association of the C-terminal domains of Tah1 and Pih1 (see Figure 2b). 

The TP complex also binds to R2 (Rvb1-Rvb2) to form the R2TP complex. The R2TP complex was initially identified by a genome-wide study of Hsp90 interacting proteins in budding yeast [96]. The R2 proteins have been found in several chromatin complexes such as the histone acetyltransferase Tip60, chromatin remodeling complexes Ino80 and SWR-C, telomerase, and the ASTRA complex [39]. Due to their presence in various complexes and their discovery by different approaches [97], Rvb1 and Rvb2 are also known as Pontin/Reptin, TIP49/48, RuvBL1/RuvBL2, and ECP54/51. Here, we refer to the two proteins as Rvb1 and Rvb2. Rvb1 and Rvb2 are similar and are paralogs that share moderate sequence similarity to the bacterial holiday junction resolvase RuvB [98,99]. The two proteins are highly conserved in eukaryotes and essential for viability in all known model organisms. They are both in the AAA+ family of ATPases that bind and hydrolyze ATP, and as is the case with RuvB, the R2 complex has a low level of 5′-3′ DNA helicase activity [100,101,102,103]. The AAA+ core of R2 (domain 1 and domain 3 or DI and DIII) form a single or double ring of heterohexamer of the two proteins alternating in position around the ring [104,105,106,107]. The insertional domain 2 (DII) of ~100 amino acids in Rvb1 and Rvb2 protrudes outward from the ring that binds to other proteins, such as the TP complex [108] and single-strand RNA/DNA or double-strand DNA [109]. In the double ring of the dodecamer, the DII domains form the interface between the two hexamers. In solution, R2 exists as the single-ring hexamer as well as the double-ring dodecamer [110,111]. In the presence of TP, R2 mainly exists as a hexamer, suggesting TP interrupts the double ring structure and drives the equilibration towards the single ring hexamer. The cryo-EM structure of yeast R2TP shows that the R2 hexamer ring with the DII domains forms an open basket that accommodates a single copy of TP as a lid [108,110]. TP binds to R2 by multiple contacts with the DII domains. The binding of TP (and potentially its interacting proteins) to the R2 hexamer is sensitive to the binding or hydrolysis of nucleotides (ADP and ATP) as the nucleotides dissociate TP from R2, likely via a conformational change in the DII domains of R2TP complex. 

The Tah1 homolog in humans is RPAP3, which is six times larger than yeast Tah1. RPAP3 has two TPR domains in the N-terminus and a RUVBL1 (Rvb1) binding domain RBD in the C-terminus. The cryo-EM structure of human R2TP shows that RPAP3 binds to the R2 hexamer ring, mainly via the RBD domain on the ATPase side, not the DII side [112]. The TPR domains of RPAP3 are mobile and invisible. A small region of ~20 amino acids in the linker region of RPAP3 binds to the CS domain of PIH1D1 (Pih1), which binds to the DII side of the R2 hexamer in the same manner as the yeast Pih1 in the R2TP complex. Interestingly, human TTT can directly bind to the R2 hexamer by competing with the RPAP3-PIH1D1 complex [12]. The concave inner surface of Tti1 wraps around the protruding DII domain of a RUVBL1 subunit. The DII domain of the adjacent RUVBL2 subunit fits into the cleft formed at the juncture of the Tti1 C-terminus and the N-terminus of Tti2, interacting with the helices from the convex faces of both proteins (Figure 2a). The TTT binding suppresses the ATPase activity of the R2 hexamer, suggesting that TTT may modulate the nucleotide binding and hydrolysis by the R2 hexamer such as the TP complex in yeast [12]. Subsequent biochemical and structural analyses showed that the competitive binding of TTT to the DII domains of the R2 hexamer is conserved in yeast [12]. Furthermore, the budding yeast R2-TTT complex can pull down the *Kluyveromyces marxianus* TOR1-Lst8 complex. The kmTOR1 has an 83% sequence similarity to the budding yeast TOR2. Interestingly, the kinase activity of kmTOR1 is not suppressed by the yeast R2-TTT, suggesting that the yeast R2-TTT can still bind to the active form of kmTOR1, and the binding does not prevent kmTOR1 from accessing its substrates nor convert it to an inactive state. 

## 6. TTT Promotes the Co-Translational Maturation of PIKKs

An early study has shown that in human cells, TTT exists sub-stoichiometrically with PIKKs, suggesting a turnover of TTT for the maturation of PIKKs [13]. Furthermore, TTT interacts with the newly synthesized peptides, not the matured PIKK proteins [13]. Since the TTT physically interacts with Hsp90 and chemical inhibition of Hsp90 suppresses the binding of TTT with PIKKs and reduces the levels of PIKKs [13,48], it is proposed that the TTT works with Hsp90 as a co-chaperone for biogenesis of PIKKs. However, the exact mechanisms remained elusive. Two general mechanisms have been proposed for the proper folding and assembly of protein complexes to minimize the misfolding and subsequent proteotoxic stress [113,114]. One mechanism involves pleiotropic chaperones that assist the folding of newly synthesized subunits and proper protein–protein interactions for complex formation. The other is through co-translational interactions between subunits of the same complex that drives protein maturation and complex assembly [113,115]. The assembly of some large multimer complexes such as proteasome, however, may take advantage of both mechanisms [116]. 

A recent study has shown that as is the case with mammals and budding yeast [117], the TTT promotes the assembly of Tra1 and Tra2, the two kinase-inactive TRRAP paralogs in fission yeast, into the highly conserved transcriptional co-factor SAGA and NuA4 complexes, respectively [118]. This complex assembly process, however, does not occur at the gene promoters in the nucleus because Tel2 is not enriched at the SAGA- and NuA4-bound promoters [10]. Consistent with this, Tel2 is mainly found in the cytoplasm in both fission yeast and mammalian cells [10,60]. A more recent study using RNA immunoprecipitation has found that the TTT is enriched with the mRNAs of all six PIKKs in fission yeast, showing that the TTT promotes the PIKK maturation co-translationally [10] (see Figure 3). TTT does not bind to the N-terminal HEAT repeats newly synthesized by the ribosome that share less sequence similarity among the PIKKs. It binds to the FAT and the kinase domains or the FATKIN domain, indicating that the TTT interacts with the newly synthesized peptides only when the more conserved FATKIN comes out of the ribosome. The short and highly conserved FATC is required for the biological functions of the PIKKs [117,119]. This recent study also found that consistent with the essential function of the FATC, the last two conserved hydrophobic amino acids of PIKKs are the key residues for the final maturation step of the newly synthesized PIKKs, which confirms previous studies in budding yeast [43,117,119]. Most of the PIKKs are multimeric complexes. Interestingly, their binding partners are not involved in the TTT-promoted PIKK maturation process, as TTT shields the interacting surfaces of the PIKK binding partners. It is thus proposed that the complex formation of PIKKs with their binding partner occurs only after the newly synthesized PIKKs have been properly folded by the TTT co-chaperone [10]. Although this study also found that Asa1 works with the TTT complex as in the case of budding yeast [39,42], it remains unclear how the TTT interacts with Hsp90 for PIKK maturation because, as mentioned above, the homologs of Pih1 and Tah1 still remain to be uncovered in fission yeast [10]. 

## 7. Other Functions of the TTT Complex in Genome Maintenance

### 7.1. TTT Specifically Regulates the Stability of a Subset of PIKKs via CK2 Phosphorylation

In addition to the co-translational maturation of all PIKKs, studies have shown that some PIKKs can be selectively regulated by the TTT. As mentioned above, Tel2 phosphorylation by CK2 promotes the association of the TTT with R2TP-Hsp90 for the maturation of PIKKs. However, the phosphosite mutations of serine^487^ and serine^491^ in Tel2 significantly reduces the protein levels of mTOR and SMG1, not the other PIKKs in human cells [60], which suggests that SMG1 and mTOR are selectively regulated by the CK2 phosphorylation of Tel2 in the TTT complex. The underlying mechanism, however, is not yet fully understood.

When DNA is damaged, the checkpoint signaling is initiated by ATM, DNA-PKcs, or ATR to promote DNA repair and cell cycle arrest. In the presence of prolonged or irreparable DNA damage, p53 is activated by the three PIKKs to promote the transcription of a set of genes for initiating the apoptotic cell death program [120]. While ATR can sense various types of DNA damage that involve ssDNA, ATM and DNA-PKcs are mainly activated by double-strand DNA breaks [53,121]. Early studies have shown that overexpression of IP6K2, a member of the inositol hexakisphosphate kinase family that produces 5-diphosphoinositol pentakisphosphate (IP7), can promote p53-associated apoptosis [122,123]. A detailed mechanistic study has shown that Tti1 in TTT can bind to IP6K2 to increase the local concentration of IP7 [87]. The increased IP7 specifically stimulates the kinase activity of CK2, which phosphorylates Tel2 at serine^485^ and Tti1 at serine^828^ and thus enhances the TTT function as the co-chaperone to specifically stabilize DNA-PKcs and ATM, not other PIKKs. Increased activities of DNA-PKcs and ATM promotes p53 phosphorylation and activation, leading to the apoptotic cell death [87]. A similar CK2-dependent mechanism for stabilization of the MRN (Mre11-Rad50-Nbs1) complex, the activator of ATM, has also been reported [124]. Unlike ATR, mTOR, SMG1, and TRRAP that are assembled with other subunits into multimer complexes, ATM and DNA-PKcs bind to their cognate activator complex MRN and Ku70/80, respectively, only after double-strand DNA is cleaved and at the damage site. It is hypothesized that the TTT may remain bound to the matured ATM, DNA-PKcs, and IP6K2 for their stability under physiological conditions. When DNA is cleaved, the TTT complex bound with ATM and DNA-PKcs is replaced by the activators at the site of DNA damage to initiate the kinase signaling. Consistent with this possibility, purified yeast TTT-R2 complex interacts with the active form of kmTOR in vitro and the interaction does not affect the kinase activity of mTOR [12]. 

In response to growth factor withdrawal or growth factor deprivation, CK2 translocates to the cytoplasm to phosphorylate serine^485^ on Tel2 and serine^828^ in Tti1 in mammalian cells [47,49]. Phosphorylated Tel2 and Tti1 within mTORC1 is targeted by the SCF^Fbox9^ ubiquitin ligase for the degradation and subsequent disassembly of mTORC1. This attenuates mTORC1 signaling while sustaining the mTORC2 function for cell growth [47]. In support of this mechanism, SCF^Fbox9^ is overexpressed in 30% of a cohort of 180 myeloma patients, which drives toward constitutive activation of the PI3K/mTOR2/Akt pathway to promote survival. 

### 7.2. TTT May Regulate the DNA Replication Checkpoint

Several lines of evidence in multiple model systems have shown that the TTT may directly regulate the kinase activities of ATR at the perturbed replication fork. In fission yeast, a genetic screen has identified a *tel2-C307Y* mutant with hypersensitivity to the replication stress induced by HU. The mutation destabilizes the TTT, leading to moderately reduced levels of Rad3(ATR) and Tel1(ATM) and short telomeres [35]. The remaining Rad3 (~60%) is still active because its kinase activity is largely intact in the DNA damage checkpoint and is required to promote the cell growth of the mutant under physiological conditions. However, the Rad3 signaling in the replication checkpoint is eliminated in the *tel2-C307Y* mutant [35]. When Tel2 was acutely depleted in fission yeast, the major defect is also the Rad3-activated replication checkpoint, not the Rad3-activated damage checkpoint [36]. These results imply that in fission yeast, Tel2 or the TTT may participate in the replication checkpoint to facilitate the Rad3 signaling. Similar results have also been observed by some early studies in mammalian cells that Tel2 depletion mainly causes the replication checkpoint defect. For example, it has been shown that human Tel2 interacts with FANCM and FAAP24 proteins [32,125], which target the FA (Fanconi anemia) core complex to chromatin under replication stress for proper ATR signaling in the replication checkpoint [33]. Human Tel2 also forms a complex with ATR-ATRIP and promotes the interaction between ATRIP and its activator TopBP1 for the replication checkpoint [34]. Human Tel2 has also been found to interact with LARG(Rho-guanine nucleotide exchange factor) protein, and depletion of LARG causes a replication checkpoint defect [126]. In budding yeast, although the *tel2-1* mutant described above has moderately reduced Tel1(ATM) and Mec1(ATR), the mutation eliminates specifically the Tel1 kinase signaling at the sites of strand breaks. Tel2 physically interacts with both Tel1 and Mec1, and the interaction with Tel1 is specifically required for Tel1 localization to the site of strand breaks. Interestingly, up-regulation of Tel1 to wild-type or even a higher level in the *tel2-1* mutant did not fully rescue the Tel1 localization and signaling. Although the up-regulated Tel1 may not be fully matured in the *tel2-1* cells, these results imply that in budding yeast, Tel2 may function to facilitate Tel1 localization to DNA strand breaks and, thus, its phospho-signaling to the downstream targets [30,31]. Collectively, these previous results show that Tel2 may contribute to a specific checkpoint pathway in both yeasts and mammalian cells. Although these studies did not examine Tti1 and Tti2, it is likely that Tel2 mediates the cellular functions as a component of the TTT. 

However, some previous studies are inconsistent with the checkpoint function of the TTT. For example, although Tel2 and Tti2 are distributed throughout the budding yeast [42,117] and mammalian cell [41], an early study in mammalian cells showed that Tel2 is mainly found in the cytoplasm, not the nucleus [60]. Another study in mammalian cells has shown that while Tti1 is mainly distributed in the cytoplasm, Tel2 shuttles between the nucleus and cytoplasm. In the presence of serum starvation, phosphorylation of Tel2 by CK2 locks Tel2 in the cytoplasm [47]. These suggest that the distribution of Tel2 or the TTT is a dynamic process that can be regulated by CK2 phosphorylation. In *C. elegans*, chromatin recruitment of ATL-1(ATR) remains intact in the HU-treated *rad-5* (*tel2*) mutant [25]. In mammalian cells, preliminary data have shown that Tti1 and Tel2 do not localize to sites of DNA damage [40,41]. Whether Tel2 or the TTT participates in the checkpoint pathways needs further investigation.

## 8. TTT in Disease and Cancer Chemotherapy

Unlike the pleiotropic phenotypes of *tel2* mutants in the model organisms described in the early studies, non-lethal mutations in any one of the TTT components mainly cause intellectual disability and neuronal and physical deformities in humans, which is consistent with their interdependency as a single functional unit. Individuals with compound heterozygous of several rare variants such as C367F, D720V, and R609H in Tel2 have been linked to the You-Hoover-Fong syndrome characterized by intellectual disability, global developmental delay, dysmorphic facial features, microcephaly, abnormal movements, and abnormal auditory and visual function [14,15,127,128,129]. Missense mutations within human Tti2 cause an autosomal recessive disorder defined by intellectual disability, microcephaly, short stature, behavioral problems, skeletal abnormalities, and facial dysmorphic features [16,130,131]. Individuals who carry bi-allelic or homozygous Tti1 variants show neurogenetic disorders with intellectual disability, microcephaly, and brain malformations [17,132]. Some of these variants or mutations have been found to destabilize the TTT complex, leading to reduced protein levels and the activities of the PIKKs, which may explain the neurodevelopmental disorders [14,17]. The disease phenotypes caused by the pathogenic biallelic variants in the TTT are different from those by mutations in ATR (Seckel syndrome), ATM (ataxia telangiectasia), and DNA-PKcs (immunodeficiency) but share some similarities with the Smith-Kingsmore syndrome caused by dominant missense mutations in mTOR. This suggests that although the disease mechanism is understandably complex, misregulation of mTOR may play a major role in the neurodevelopmental disorders caused by the TTT mutations [133,134]. 

In addition to the neurodevelopmental disorders and the cancer predisposition syndromes caused by mutations in ATR, ATM, mTOR, and their regulated pathways, studies have shown that mutations in human Tel2 may also contribute to the development of sessile serrated adenomas and breast cancer [135,136]. As mentioned above, Tti1 binds to IP6K2 and participates in the signaling pathway of the p53-associated apoptotic cell death pathway [87]. The defect of this pathway may promote tumor survival by avoiding apoptotic cell death. In myeloma, Tel2 and Tti1 in mTORC1 are targeted by SCF^box9^ ubiquitin ligase for their degradation, which inactivates mTORC1 and thus promotes mTORC2-dependent survival [47]. Furthermore, overexpression of Tti1 has been shown to promote colorectal and non-small-cell lung cancer via the mis-regulation of the mTOR pathway [137,138]. Therefore, targeting the TTT complex may provide a new strategy for cancer chemotherapy.

Indeed, because of its essential functions, the TTT and its associated proteins such as Hsp90, CK2, and R2TP complex have been investigated as the targets for developing new cancer chemotherapeutics. During the past decades, a series of small molecule inhibitors of Hsp90 and CK2 have been developed, and most of them are still in clinical trials (see [139,140,141,142] for the reviews). Toxicity and poor bioavailability prevent them from being used in cancer patients. Small molecule inhibitors have also been developed for R2 complex that have the potential for clinical trials [143,144]. Hsp90, CK2, and R2 are enzymes, which make them druggable targets. On the contrary, the TTT does not have any enzymatic activity and all three subunits are predominantly α-solenoids, which makes the complex less druggable by small molecule inhibitors. Surprisingly, a recent study has shown that ivermectin (IVM), a macrolide antiparasitic antibiotic, binds to the C-terminal domain of human Tel2 and thus suppresses the TTT functions, leading to reduced activities of PIKKs, particularly the mTOR activities [18]. IVM has been approved by the FDA since 1987 for successful use by millions of patients with river blindness, elephantiasis, and scabies [145,146]. Because of this contribution, the discoverers of IVM, Satoshi Ömura and William C. Campbell, won the Nobel Prize in Physiology or Medicine in 2015. In addition to the remarkable antiparasitic efficacy, IVM has been shown to have antiviral [147,148,149] and strong anticancer activities [150,151], and it has also been shown that IVM can reverse multiple drug resistance in tumor [152]. Since the safety has been tested by millions of patients, IVM is currently being repurposed for cancer treatment [146,150]. Although the *tel2-K749T* mutation abolishes the binding of IVM [18], a structural study in the future will likely reveal more details of the drug-protein interactions, guiding the derivatization of IVM for enhanced therapeutic efficacies. Since Tel2 inhibition also reduces the activities of ATR, ATM, and DNA-PKcs, IVM may also be used in combination with the anticancer drugs such as bleomycin and cisplatin that damage DNA. Nonetheless, this surprising discovery has validated the TTT as a druggable target for the treatment of cancer or other diseases. 

## 9. Summary and Future Perspectives

The discoveries of Tel2 in the 1980s in yeast and *C. elegans* and by different screening methods suggest that it works in distinctive pathways such as genome stability, telomere maintenance, and life span. The complex phenotypes of *tel2* mutants were reconciled only until 2007 by the seminal discovery of the de Lange’s group that Tel2 functions as a co-chaperone for stability of all six PIKKs in human cells [13,41]. Since then, this mechanism has been verified and subsequently enhanced by other laboratories. First, Tel2 does not work alone, it works within the heterotrimeric TTT complex as a single functional unit [11,12,37,39,40]. Second, the TTT interacts with molecular chaperones Hsp90 and Hsp70 [13,40] via CK2 phosphorylation of Tel2 and the R2TP complex [42,60,61]. Third, chemical inhibition of Hsp90 destabilizes and negatively regulates the PIKKs [13,48]. Fourth, TTT is enriched with the mRNAs of all PIKKs in fission yeast, and, therefore, it promotes PIKK maturation co-translationally [10]. Finally, chemical inhibition of Tel2 by IVM negatively regulates ATR, ATM, mTOR, and DNA-PKcs [18]. 

Although this mechanism has been verified by many laboratories and explains the pleiotropic phenotypes of *tel2* mutants, some previous studies have shown inconsistent results. For example, SMG1 and mTOR are significantly more sensitive than other PIKKs to the Tel2 mutations of two CK2 phosphorylation sites [60]. CK2 phosphorylation of Tel2 and Tti1 can also specifically disrupt the mTORC1, and not the mTORC2 complex, to promote survival after growth factor withdrawal or in tumor cells [47]. Inositol pyrophosphate IP7 promotes apoptosis via specific upregulation of ATM and DNA-PKcs by CK2 phosphorylation of Tel2 and Tti1 in the TTT [87]. In fission yeast, mutation or acute depletion of Tel2 mainly affect the Rad3(ATR)-activated replication checkpoint, not the Rad3-activated DNA damage checkpoint [35,36]. In budding yeast, the *tel2-1* mutation specifically affects Tel1(ATM) signaling at the site of DNA strand breaks [30]. Furthermore, fission yeast lacks the homologs of Tah1 and Pih1, and budding yeast may stabilize its PIKKs through two separate TTT-dependent pathways [42]. While TTT works in the cytoplasm for the co-translational PIKK maturation, Rvb1 and Rvb2 are mainly found in the nucleoplasm [39]. Whether Rvb1 and Rvb2 contain helicase activity has been controversial and remains to be confirmed, though [109,153]. Further investigations are needed to provide new insight into this biologically important protein complex. 

As mentioned above, small molecule inhibitors of Hsp90, CK2, and R2 have been uncovered with the potential to be used in clinics for the treatment of cancer or other diseases. However, these enzymes have multiple functions. Their inhibition may have a broad cellular impact, causing side effects or toxicities that prevent further development to be used in clinics. Uncovering the mechanistic details of the TTT as well as its specific regulation of some but not all PIKKs may reveal new strategies to enhance the efficacies or reduce the toxicities of the therapeutics that target the TTT and its interacting proteins. The discovery of the Noble Prize winning medicine IVM as the inhibitor of Tel2 [18] will likely drive a new wave of research into the TTT complex for the benefits of patients with cancer or other diseases. 

## Figures and Tables

**Figure 1 ijms-24-08268-f001:**
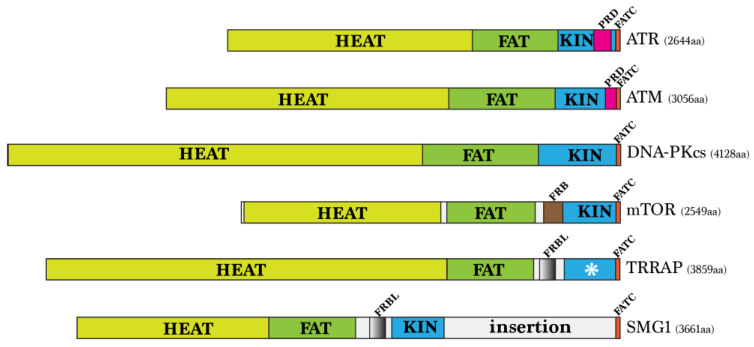
Domain organization of human PIKKs. The relative position of HEAT repeat, FAT, and kinase (KIN) domains are indicated in yellow, green, and blue, respectively. PRD, the PIKK regulatory domain in ATR and ATM, is shown in pink. FRB, the FKBP-rapamycin binding domain in mTOR, is shown in brown. FRBL, the FRB-like domain in TRRAP and SMG1 is shown in steel black. Among the PIKKs, only TRRAP lacks the kinase activity, as indicated by the asterisk in its kinase domain. The function of the insertional region in SMG1 remains undefined. This diagram was illustrated based on references [56,57,58,59].

**Figure 2 ijms-24-08268-f002:**
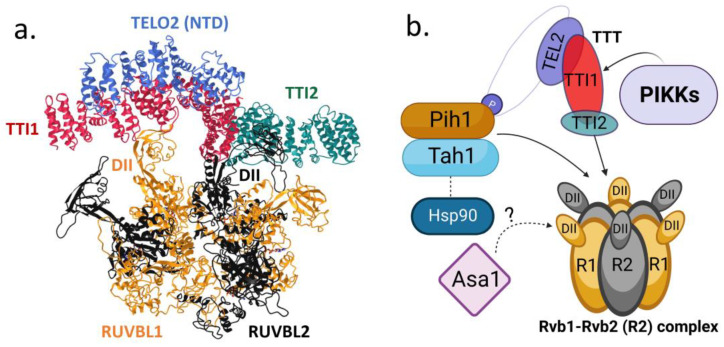
Structure of human TTT complex and its interactions with R2 (RUVBL1-RUVBL2), TP (Tah1-Pih1), Hsp90, and Asa1 for stability of PIKKs. (**a**) Cryo-EM structure of human TTT-R2 complex [12] is shown by using 3D structure viewer iCn3D. The single R2 heterohexameric ring is composed of the DI and DIII domains of RUVBL1 and RUVBL2. The TTT complex binds the R2 complex by interacting with the DII domains of RUVBL1 and RUVBL2. (**b**) The phosphorylation dependent interactions of the TTT with the TP complex, which brings the molecular chaperone Hsp90 in proximity via the R2 complex for promoting the stability of PIKKs. Asa1 may also contribute to the TTT-dependent stability of PIKKs (see text for details).

**Figure 3 ijms-24-08268-f003:**
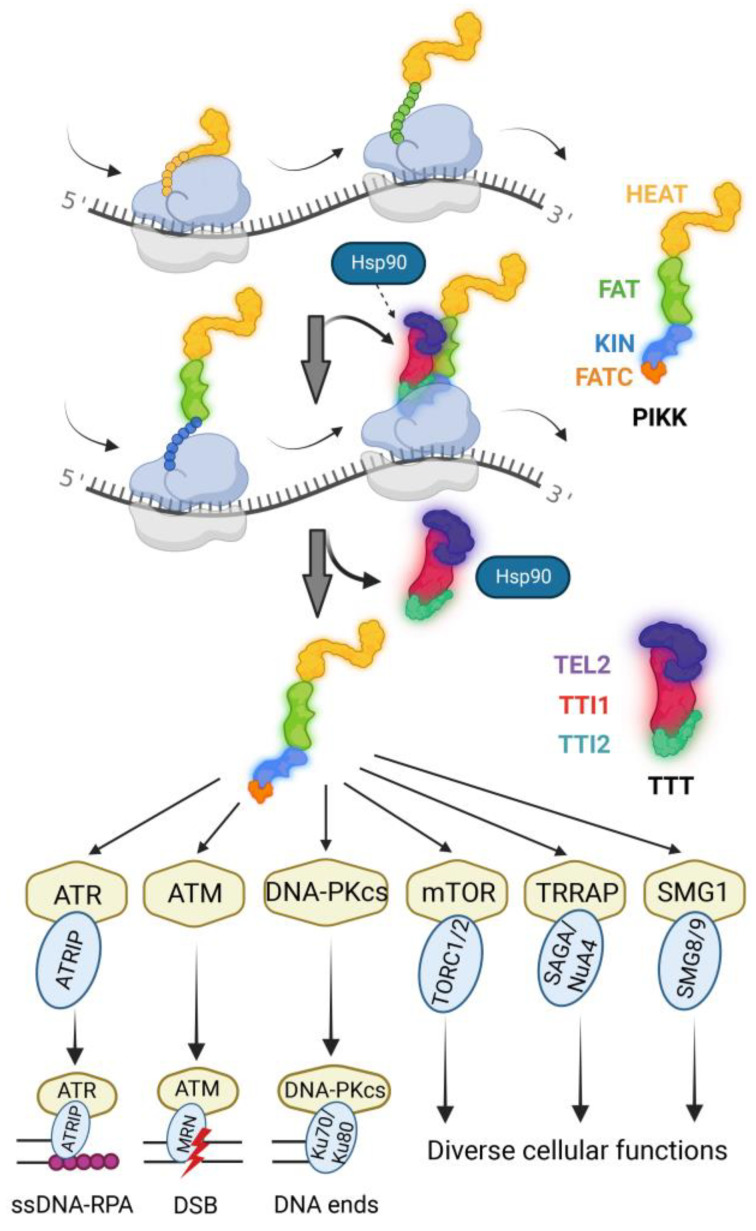
Co-translational maturation of PIKKs by the TTT complex. The diagram is inspired by the work done by Toullec et al. [10].

**Table 1 ijms-24-08268-t001:** Conserved proteins and complexes in eukaryotes. Conserved proteins or complexes are colored in groups. Undefined or unidentified proteins are indicated by question marks.

Complex	Human	Budding Yeast	Fission Yeast	*C. elegans*
	TELO2	Tel2	Tel2	CLK-2/RAD-5
**TTT**	TTI1	Tti1	Tti1	R10H10.7
	TTI2	Tti2	Tti2	C28H8.3?
**Hsp90**	HSP90α/HSP90β	Hsp82/Hsc82	Hsp90	HSP-90
**R2**	RUVBL1	Rvb1	Rvb1	RUVB-1
	RUVBL2	Rvb2	Rvb2	RUVB-2
**TP**	RPAP3	Tah1	?	RPAP-3
	PIH1D1	Pih1	?	?
**Asa1**	GNB1L	Asa1	Asa1	?
	ATR	Mec1	Rad3	ATL-1
	ATM	Tel1	Tel1	ATM-1
**PIKKs**	DNA-PKcs	?	?	?
	mTOR	Tor1, Tor2	Tor1, Tor2	LET-363
	SMG1	?	?	SMG-1
	TRRAP	Tra1, Tra2	Tra1, Tra2	TRR-1
**ATRIP**	ATRIP	Ddc2	Rad26	?
	MRE11	Mre11	Mre11	MRE-11
**MRN**	RAD50	Rad50	Rad50	RAD-50
	NBS1	Xrs2	Nbs1	NBS-1
**KU70/80**	KU70/80	Ku70/Ku80	Pku70/Pku80	CKU-70/CKU-80

## Data Availability

Not applicable.

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
