# Peer review of "TTT (Tel2-Tti1-Tti2) Complex, the Co-Chaperone of PIKKs and a Potential Target for Cancer Chemotherapy"

_ijms, 2023, doi:10.3390/ijms24098268_

Round 1
Reviewer 1 Report
The manuscript provides a comprehensive review about the function of the TTT (Tel2-Tti1-Tti2) complex in regulating PPIKs folding/stability as a co-chaperone. The TTT complex binds to Hsp90 through the R2TP (Rvb1-Rvb2-Tah1-Pih1) complex for co-translational maturation of the PIKKs, including ATR, ATM, DNA-PKcs, mTOR, SMG1, and TRRAP, which are critical for cell signaling and DNA damage response. This topic is interesting and the manuscript is well-written. Some minor issues need to be addressed before publication.
Line 117: The reviewer has a question about this title. It seems that the direct role of TTT is to promote the correct folding of PPIKs, thus the stability of PIKKs is secondary to the folding.
The main point from line 403-409 is that SMG1 and mTOR are specifically regulated by CKII phosphorylation, but the next paragraph discussed that CKII phosphorylates Tel2 to regulate the stability of DNA-PK and ATM. This causes confusion.
Line 459-466: It seems that all the examples here are TTT-independent Tel2 functions. Is it true? This needs to be clarified.
One of the main focus of this review is that TTT could be a therapeutic target for cancers, but the explanation of the role of TTT in cancer development is not sufficient (line 515-524).
Line 45: phospho-signaling
Line 45: A brief introduction about the TRRAP is needed here.
Line 61-64: The link between Tel2 inhibitors and anti-cancer activity is missing here
Line 71: rad-5 mutant
Line 72: tel2-1 mutant
Line 93-94: uncover two Tel2 interacting proteins
Line 180: et al.
Line 182: Using phospho-peptides
Line 210: “by testing their ability to resist local unfolding” This part needs to be reworded.
Line 225: It would be better to change to “…as a component of R2TP, Pih1 recruits…”
Line 399: Hsp90
Line 430-432: This sentence is hard to follow.
Line 532: Hsp90
Line 540: …discoverers of IVM by Satoshi Omura…
Author Response
Response to the comments by reviewer #1
Note: reviewers’ comments are colored blue and the answers are in black.
The manuscript provides a comprehensive review about the function of the TTT (Tel2-Tti1-Tti2) complex in regulating PPIKs folding/stability as a co-chaperone. The TTT complex binds to Hsp90 through the R2TP (Rvb1-Rvb2-Tah1-Pih1) complex for co-translational maturation of the PIKKs, including ATR, ATM, DNA-PKcs, mTOR, SMG1, and TRRAP, which are critical for cell signaling and DNA damage response. This topic is interesting and the manuscript is well-written. Some minor issues need to be addressed before publication.
Line 117: The reviewer has a question about this title. It seems that the direct role of TTT is to promote the correct folding of PPIKs, thus the stability of PIKKs is secondary to the folding.
Response: We appreciate the suggestion and change the subtitle to “Tel2 deletion reduces the protein levels of all PIKKs”. See line 118 in the revised manuscript with mark-ups.
The main point from line 403-409 is that SMG1 and mTOR are specifically regulated by CKII phosphorylation, but the next paragraph discussed that CKII phosphorylates Tel2 to regulate the stability of DNA-PK and ATM. This causes confusion.
Response: Yes, the results with CK2 phosphorylation appears to be somewhat inconsistent and needs more detailed investigation. One of the possible explanations is that the phosphorylation sites are different in these studies. One study identified the CK2 phosphorylation sites on Tel2 to be serine487 and serine491, while the other studies focus on serine485 in Tel2 and serine828 on Tti1. It is also possible that different cell lines were used in those studies. Nevertheless, we added the phosphor-sites in the revised manuscript to mitigate the confusion, see lines 434-466.
Line 459-466: It seems that all the examples here are TTT-independent Tel2 functions. Is it true? This needs to be clarified.
Response: Most of the early studies focus on Tel2, particular before Tti1 and Tti2 were discovered. So far, several lines of evidence have shown that the TTT functions as a single unit, see summary. Therefore, it is likely that the functions observed in tel2 mutants or by Tel2 depletion in the previous studies are related to the TTT, not by Tel2 itself, although further experiments are needed to confirm this possibility. See lines 513-514.
One of the main focus of this review is that TTT could be a therapeutic target for cancers, but the explanation of the role of TTT in cancer development is not sufficient (line 515-524).
Response: We agree with the reviewer and rewrote/reorganized this section, emphasizing on cancer. See lines 557-568.
Line 45: phospho-signaling
Response: fixed.
Line 45: A brief introduction about the TRRAP is needed here.
Response: yes, added. See line 57.
Line 61-64: The link between Tel2 inhibitors and anti-cancer activity is missing here
Response: more details are provided for the missing link here. See lines 75-76.
Line 71: rad-5 mutant
Response: fixed
Line 72: tel2-1 mutant
Response: fixed
Line 93-94: uncover two Tel2 interacting proteins
Response: fixed
Line 180: et al.
Response: fixed
Line 182: Using phospho-peptides
Response: fixed
Line 210: “by testing their ability to resist local unfolding” This part needs to be reworded.
Response: Yes, reworded. See lines 234-237
Line 225: It would be better to change to “…as a component of R2TP, Pih1 recruits…”
Response: Yes, changed accordingly. See line 251
Line 399: Hsp90
Response: fixed
Line 430-432: This sentence is hard to follow.
Response: the sentence is rewritten. See line 463-464.
Line 532: Hsp90
Response: fixed
Line 540: …discoverers of IVM by Satoshi Omura…
Response: fixed
Reviewer 2 Report
This review article is well-written. A very extensive and detailed description of the research on TTT.
I have two comments about the title: 1. In the title, please indicate which indication TTT is a therapeutic target for. For example, cancer. 2. I also recommend revising the title clearer. The current title is too broad. The conclusion and references are appropriate.
The quality of English is high.
Author Response
Response to the comments by reviewer #2
This review article is well-written. A very extensive and detailed description of the research on TTT.
I have two comments about the title: 1. In the title, please indicate which indication TTT is a therapeutic target for. For example, cancer. 2. I also recommend revising the title clearer. The current title is too broad. The conclusion and references are appropriate.
Response: We appreciate the reviewer’s suggestion and have changed the title as suggested by focusing on cancer chemotherapy.